# Exploring Marine-Derived Compounds: In Silico Discovery of Selective Ketohexokinase (KHK) Inhibitors for Metabolic Disease Therapy

**DOI:** 10.3390/md22100455

**Published:** 2024-10-03

**Authors:** Mansour S. Alturki

**Affiliations:** Department of Pharmaceutical Chemistry, College of Clinical Pharmacy, Imam Abdulrahman Bin Faisal University, P.O. Box 1982, Dammam 31441, Saudi Arabia; msalturki@iau.edu.sa

**Keywords:** ketohexokinase, marine compounds, metabolic diseases, molecular simulation, PF-06835919

## Abstract

The increasing prevalence of metabolic diseases, including nonalcoholic fatty liver disease (NAFLD), obesity, and type 2 diabetes, poses significant global health challenges. Ketohexokinase (KHK), an enzyme crucial in fructose metabolism, is a potential therapeutic target due to its role in these conditions. This study focused on the discovery of selective KHK inhibitors using in silico methods. We employed structure-based drug design (SBDD) and ligand-based drug design (LBDD) approaches, beginning with molecular docking to identify promising compounds, followed by induced-fit docking (IFD), molecular mechanics generalized Born and surface area continuum solvation (MM-GBSA), and molecular dynamics (MD) simulations to validate binding affinities. Additionally, shape-based screening was conducted to assess structural similarities. The findings highlight several potential inhibitors with favorable ADMET profiles, offering promising candidates for further development in the treatment of fructose-related metabolic disorders.

## 1. Introduction

The increasing prevalence of metabolic diseases such as nonalcoholic fatty liver disease (NAFLD), obesity, and type 2 diabetes have resulted in widespread epidemics significantly affecting global health, [1] reducing life expectancy, and diminishing the quality of life for people worldwide [2,3,4]. Over the past few decades, numerous studies have concentrated on investigating the impact of fructose, a prevalent sugar in plant-based foods and processed energy-dense foods, such as sugar-sweetened beverages, on metabolic diseases [5,6,7]. Ketohexokinase (KHK) plays a crucial role in fructose metabolism by phosphorylating fructose to fructose-1-phosphate. This enzyme exists in two isoforms, KHK-A and KHK-C, with KHK-C being more active in the liver and having a higher capacity for fructose phosphorylation [8,9,10]. Fructose metabolism contributes directly to the development of cardiometabolic diseases due to the rapid action of KHK-C, which lacks feedback inhibition. This rapid metabolism can deplete hepatic ATP after fructose consumption [11], leading to increased lipogenesis and promoting the expression of lipogenic genes [12,13]. Fructose also enhances glucose-stimulated insulin secretion [14,15,16], resulting in hyperinsulinemia and insulin resistance, critical factors in diseases like NAFLD (Figure 1) [17]. Consequently, health organizations recommend limiting daily fructose intake to mitigate these effects [18,19]. However, the widespread presence of fructose in a diet suggests that additional measures may be necessary to counteract its harmful impacts. The inhibition of KHK, particularly KHK-C, is considered a potential strategy for mitigating the adverse effects of excessive fructose consumption, which is linked to these metabolic disorders.

KHK is a member of the ribokinase superfamily and typically functions as a homodimer [20]. Each monomer consists of two domains: an N-terminal and a C-terminal domain [21]. The enzyme’s active site is located at the interface of these domains, and its structure has been elucidated through several crystal structures. One of the key structural features of KHK is its pseudo-homodimeric form, where one monomer adopts a closed conformation while the other remains in an open conformation [22,23]. This conformation is stabilized by a β-clasp interaction, which plays a critical role in the enzyme’s function and has been a target for small-molecule inhibitor design [24]. The active site of KHK binds ATP and fructose, facilitating the transfer of a phosphate group from ATP to fructose. Inhibitors targeting KHK typically interact with the ATP-binding site or the fructose-binding site. The structural insights gained from crystallographic studies have been instrumental in the rational design of potent KHK inhibitors. For example, the binding of AMP-PNP (a non-hydrolyzable ATP analog) to KHK has provided valuable information on the key interactions necessary for inhibitor binding.

Potent pyrimidine and indazole derivatives against KHK were reported through high-throughput screening and fragment-based drug design [23,25,26,27,28]. These compounds were further optimized via structure-based drug design, revealing fundamental interactions within KHK’s active site [25]. X-ray crystallography confirmed their binding, with some inhibitors achieving submicromolar potency and significant inhibitory activity against KHK [25]. Recent advances in KHK inhibitor development have focused on novel structural motifs and modifications to known inhibitor classes. Researchers have explored various chemical scaffolds, including pyrimidine/cyanopyridine [29], sulfinates [30], and saturated monocyclic heterocycles [31]; some have demonstrated good bioavailability and good pharmacokinetic profiles in preclinical studies, making them promising candidates for further development. Despite the progress in developing small-molecule KHK inhibitors, translating KHK inhibitors into clinical success has been challenging. Several inhibitors, such as Pfizer’s PF-06835919 (Figure 2) [32], showed initial promise [33] but were discontinued after Phase II trials without the reason’s disclosure. Eli Lilly’s inhibitors LY3478045 and LY3522348 faced similar issues [34,35], demonstrating that achieving sufficient efficacy and safety profiles remains a primary challenge in clinical development.

The main hurdle to using natural products, especially those of marine origin, is their reduced availability and complicated extraction and purification procedures. One example is avarone, a promising sesquiterpene quinone, which, despite being isolated in good quantities from its original matrix, deserves further biotechnological development [36]. Additionally, due to the time and expense associated with traditional drug discovery processes, bioinformatics approaches are increasingly favored for identifying novel drug targets that are specific and selective against bacterial pathogens [37,38,39]. One primary objective of screening is to discover new chemical compounds with optimal biological activity by searching commercial and public databases [40]. For example, bioinformatics techniques have been used to identify seven metabolic pathway enzymes and non-homologous membrane proteins as promising antibacterial targets [41]. Furthermore, existing targets can be leveraged to unravel new classes of drug molecules, as exemplified by identifying hits against penicillin-binding protein 2a of Methicillin-resistant S. aureus (MRSA) [42]. Telithromycin, a third-generation ketolide antibiotic, was discovered to be an effective agent specifically targeting resistant bacterial strains [43]. Thus, molecular modeling with medicinal chemistry knowledge can facilitate the development of potential targets and promising drug candidates [44].

The present in silico study focused on exploring molecules from natural marine organisms as selective inhibitors of ketohexokinase (KHK). Two drug design strategies were employed: structure-based drug design (SBDD) and ligand-based drug design (LBDD) [44]. Initially, molecular docking was used to identify the top-docked compounds [45], followed by induced-fit docking (IFD) [46], molecular mechanics generalized Born and surface area continuum solvation (MM-GBSA) [47], and molecular dynamics simulations to validate their binding affinities [48]. A shape-based screening was also conducted to assess shape similarities [49]. ADMET studies were conducted to evaluate the pharmacokinetic properties of the potential KHK inhibitors [50]. The findings of this study have the potential to contribute significantly to the ongoing efforts to enhance the biological activity of hit compounds against ketohexokinase (KHK).

## 2. Results

### 2.1. High-Quality Protein Structure Evaluation

Relevant information on protein reliability is illustrated in Appendix A. The KHK structure (6W0Z) exhibited some issues with B-factors, suggesting a disorder of atoms from their ideal equilibrium positions. Specifically, the ARG-249 atom was missing (Appendix A), and several water molecules, including H_2_O-485, 487, 493, 495, 572, 579, 580, and 581, lacked hydrogen bond partners (Appendix A). Additionally, three isolated water molecules were detected at a minimum distance of 8.983 Å from the protein. A slight bond angle deviation was observed for ASP-114 and GLU-227. Despite these issues, the overall reliability report for 6W0Z was favorable, indicating minimal concerns. Steric clashes in the KHK enzyme structure were addressed through energy minimization. These clashes indicate high-energy conformations that can cause disruptions and lead to instability during simulations (Appendix A). However, minimization can also introduce unfavorable contacts, potentially altering the enzyme’s overall conformation. Therefore, it is essential to evaluate the enzyme before and after minimization to achieve an energy-optimized structure for accurate docking and simulation predictions. The Ramachandran plot for the pre-minimized KHK enzyme showed 93.3% residues in the favored region, while the plot for the energy-minimized enzyme showed 92.5% residues in the favored region. Notably, no residues were in disallowed regions for both the pre-minimized and minimized structures. Figure 3 depicts the Ramachandran plots for both the pre-minimized and energy-minimized KHK enzymes.

### 2.2. Docking Studies

The co-crystalline ligand was redocked into its target KHK using the same procedure and protocol applied for the six hits to validate docking. Subsequently, rigid-body superposition was performed using Maestro’s structure superposition tool to align the predicted lowest energy conformation of the target with its corresponding co-crystalline ligand (Appendix A). The classical RMSD from the co-crystalline pose was calculated for the predicted binding poses, with an RMSD < 2 Å considered an effective threshold for validating correctly posed molecules [50,51]. The results showed good binding mode superimposition, with an RMSD of 0.3004 Å for PF-06835919, reflecting the accuracy of Glide’s pose prediction (Figure 4). MM-GBSA study identified five hits, including compound **1**(CMNPD12445), compound **2**(CMNPD799), compound **3**(CMNPD24755), compound **4**(CMNPD27745), and compound **5**(CMNPD21775) (Figure 5). The binding affinities of the five hits were assessed against KHK. Initially, the inhibition profiles of these five hits were examined by docking them into the binding pockets of the target, investigating their binding patterns, target interactions, and binding affinities compared to the reference PF-06835919. 

### 2.3. Computational Analysis of the Five Hits Binding to KHK

The findings presented in Table 1 offer valuable insights into the binding affinities and interaction profiles of the selected hit compounds with the target enzyme KHK. The application of induced-fit docking (IFD) was crucial in generating accurate complex structures for these hits, allowing for the identification of true binders that might have been initially overlooked due to poor scores (false negatives). This was accomplished by employing multiple receptor conformations obtained through the IFD protocol rather than relying on a single rigid conformation, thereby enhancing the reliability of the screening process [52]. Compound **1** exhibited significant polar interactions, a high docking score, and binding affinity. (1) The carbonyl oxygen group occupancy of the ATP-ribose pocket with the oxygen atom filling a small but crucial sub-pocket defined by Phe-260 (normally filled by the methylene of the ATP ribose sidechain); (2) a hydrogen-bonding interaction between the para hydroxyl group and a conserved water that forms a hydrogen bond with the backbone CO of Cys-282; (3) the primary alcohol and the ester oxygen groups filling the hydrophobic pocket defined by the Lys-193, Ala-224, and Ala-226 residues; (4) one of the oxygen atoms of the carboxylate group was well-positioned to interact with the cationic guanidinium group of Arg-108 (distance= 2.37 Å). In addition, the backbone NH of Gly-255 and Gly-257 and (5) the aromatic core provides rigidity and enhances binding affinity by reducing the entropy loss upon binding [52]. Compound **2** exhibited comparable key amino acid interactions, a high docking score, and binding affinity to **1**; (1) an ionic bond between the oxygen atoms of the carboxylate group and Arg108, the backbone NH of Gly-255 and Gly-257; (2)two hydroxyl groups of the cyclopentane ring formed strong hydrogen bonds with Ala-226 and Gly-229, and conserved water that forms a hydrogen-bond with the backbone CO of Cys-282. Similar interactions with key amino acid residues were observed for **3**, **4**, and **5**. These interactions support their relatively high binding affinity and docking scores. Favorable binding-based 2D and 3D docking positions interact with key residues within the binding pocket, as shown in (Figure 6 and Figure 7).

### 2.4. Binding Free Energies Analysis

To validate the affinity of compounds to KHK, post-docking of MM-GBSA was performed to yield different free energies of the complexes. MM-GBSA is considered reliable for this purpose as it is more accurate than docking predictions [53]. The MM-GBSA method was used to cross-validate the docking findings. Table 2 presents the different binding free energies calculated by the MM-GBSA method. In MM/GBSA, the net binding free energy of the systems is −51.66 kcal/mol for **1**, −45.23 kcal/mol for **2**, −38.29 kcal/mol for **5**, −35.66 kcal/mol for **3**, and −31.25 kcal/mol for **4**. 

### 2.5. Molecular Dynamics Simulation of ***1*** and ***2*** Binding to the KHK Target

Based on the initial docking and MM-GBSA results, **1** and **2** were chosen for further analysis by MD perturbation. In the MD study, the KHK RMSDs were equilibrated between 30 and 100 ns and maintained below 4.0, signaling that they formed stable complexes [54] with **1** and **2** (Figure 8A,B). Interestingly, the RMSDs of **1** and **2,** in reference to the KHK backbone (Lig Fit Prot), showed that the ligands remained at the receptor binding site throughout the analysis [55] (Figure 8A,B). 

To further assess the structural components of KHK for considerable deformations during the analysis, its root mean square fluctuations (RMSFs) were considered [56]. When assessed in complexes with either **1** (Figure 9A) or **2** (Figure 9B), KHK experienced the highest oscillations (RMSF 4.2–5.2 Å) on amino acid residues between 0 and 50 and then at about 95–105, depicting that these residues were freely flexible [57]. However, none of these showed relevance in maintaining the ligand–receptor complex (Figure 10), while most of the backbone structure remained relatively rigid (Figure 9) and played an important part in interacting with ligands (Figure 10). 

The prominent interactions that significantly contributed to the established binding poses of **1** and **2** to 6W0Z included a mixture of hydrogen bonds (with or without water bridges), as well as hydrophobic, and ionic interactions (Figure 10). Consistently, Ala-226 established the highest hydrogen bond interaction with both ligands (Figure 10A,B), while Glu-227 participated mainly through the water bridge-mediated hydrogen and ionic interactions (Figure 10C,D). Further water-bridge-assisted hydrogen bonding could be seen with Ala-244, Phe-245, Thr-253, and Cys-282, involving both ligands (Figure 10A–D). The direct ionic interactions with Arg-108 were obtained for both ligands, where it was most significant for **1** (19%) than **2** (5%) (Figure 10C,D). Some hydrophobic amino acid residues, such as Pro-247 and Phe-260, were in very close proximity to induce significant hydrophobic interactions with **1**, while **2** relied on Pro-247 and Ala-285 for this interaction. In general, the MD results were in agreement with the initial docking results (Figure 6 and Table 1). 

### 2.6. Shape Similarity Prediction

Two molecules with comparable shapes are likely to fit into the same binding pocket and demonstrate similar biological activity. The similarity between two molecules is measured using various descriptions of molecular shape as previously outlined. This approach has been successfully utilized as a virtual screening tool to identify molecules from a chemical library that resemble a specific query [58]. According to Table 3, **2** exhibited the highest score for shape similarity compared to the other compounds. The structural resemblance between **2** and PF-06835919 plays an essential role in their interaction with KHK. 

### 2.7. Alanine Scanning Analysis

Specific residues were mutated to understand their functional significance involved in the docking pose and the strong interaction of potential hits with KHK. In this context, Arg-108, Gly-255, Gly-257, and Cys-282 were replaced by alanine to induce native changes in the structure without affecting the overall conformation of the enzyme. This approach showed that the hits have lower docking scores except for **3** and **4** (C282A), as seen in Table 4, compared to the initial docking results (Table 1). The decline is observed in most mutant residues owing to their importance in the KHK binding chamber, which was reported previously [32]. 

### 2.8. ADMET and Drug-Likeness

In drug development, inadequate pharmacokinetics can result in substantial drug waste and expensive costs. Assessing the ADMET (Absorption, Distribution, Metabolism, Excretion, and Toxicity) characteristics is essential during drug discovery and development. An ideal drug candidate should be effective against its intended target and demonstrate favorable ADMET profiles at therapeutic levels [59]. A comprehensive pharmacokinetic assessment was carried out on the five potential drug candidates. Molecular descriptors were employed to examine absorption mechanisms and potential administration routes to estimate bioavailability, water solubility, Caco2, and human intestinal absorption. Table 5 reveals compounds that exhibit good oral absorption, except **5**, which requires structural improvements for better water solubility and intestinal absorption. None can penetrate the BBB or CNS, indicating limited distribution via these routes. Metabolism studies show that CYP3A4 enzyme metabolizes **1**, **3**, and **4**, whereas the remaining compounds do not involve CYP450 isoforms, indicating other routes of metabolism. Their excretion profiles, assessed by total clearance and renal OCT2, suggest efficient body clearance without being substrates for renal OCT2. Pharmacodynamic properties were also inspected by analyzing toxicological descriptors such as AMES toxicity, maximum tolerated dose, hepatotoxicity, and hERG inhibition. Interestingly, **1** and **2** are predicted to show no toxicity except for **3**, **4**, and **5**, suggesting future experimental exploration. In assessing the potential drug candidates’ ADMET profiles, the drug-like profile of the inhibitors was also analyzed. The ADME characteristics of the molecules align with established drug-like criteria, including notable frameworks such as Lipinski’s rule of five [60], Ghose [61], Veber [62], Egan [63], and Muegge [64] rules. This suggests that the potential candidates will likely exhibit favorable pharmacokinetics and potentially strong oral bioavailability. Medicinal chemistry evaluations indicate that the molecules are synthetically accessible and show no indicators for pans assay interference structures (PAINS), which underscores their selective binding to KHK. Table 6 shows that **4** meets all criteria without any alerts, and all examined inhibitors generally conform to the established drug-like criteria.

## 3. Discussion

KHK is a key-limiting enzyme in fructose metabolism that catalyzes the phosphorylation of fructose into fructose-1-phosphate. The rapid metabolism of fructose via this pathway contributes to lipogenesis and ATP depletion, thereby promoting metabolic syndrome and NAFLD [8,9,10]. More recently, it has been demonstrated that KHK-C promotes endoplasmic reticulum stress, further exacerbating liver disease in both diet-induced and genetic murine models of NAFLD [65]. The importance of KHK for fructose-related metabolic diseases has promoted the development of KHK inhibitors that are now being assessed in clinical trials. One such inhibitor is PF-06835919 [32]. Excessive intake of fructose has been associated with an increased risk of cardiometabolic syndrome and cardiovascular disease [66].

Structure-based drug design (SBDD) is a strategy in drug development that utilizes the three-dimensional structure of a target protein to direct the design and optimization of potential inhibitors. It relies heavily on high-resolution structural information, usually acquired through X-ray crystallography or nuclear magnetic resonance spectroscopy, to identify key interactions within the binding site of the target protein. This involves docking small molecules into the active site to predict their binding mode and affinities, followed by iterative cycles of design and optimization toward improvement in binding strength, specificity, and overall drug-like properties. SBDD is particularly useful for optimizing the fit and orientation of potential drug candidates within the binding pocket, allowing the rational design of compounds with enhanced potency and selectivity. On the other hand, ligand-based drug design (LBDD) is used when the structure of the target protein is not available. By using the structural information from known active compounds, LBDD will make predictive models that can spot novel compounds with similar biological activities. Several of the typically applied techniques in LBDD are QSAR modeling, pharmacophore modeling, and similarity-based virtual screening. These methods analyze the chemical features and spatial arrangements critical for the biological activity of known ligands, utilizing that information for screening or designing new compounds [44].

In our study, we employed state-of-the-art in silico techniques to identify potential inhibitors binding with the active site of KHK-C, which demonstrated the efficiency of the combined use of both SBDD and LBDD approaches [54,67]. Docking provided an initial set of potential compounds, which were further validated for predicting binding affinities using IFD and MM-GBSA [68]. The Root Mean Square Deviation (RMSD) and Root Mean Square Fluctuation (RMSF) analyses were conducted to evaluate the stability and flexibility of protein–ligand complexes during simulation. The low RMSD and RMSF values show that these complexes retained their structural integrity, and the residues at the binding site displayed minimal fluctuations [48]. Molecular dynamics simulations were carried out on the two top-ranked compounds, **1** and **2**, to assess their dynamic stability and binding interaction with the active site of KHK. This step is very important in confirming the reliability and efficacy of the selected inhibitors through their observation in the simulated physiological environment. From this 100-ns MD simulation, the RMSD of both the ligands and the RMSF of the KHK target maintained values below 3.0 Å, testifying to the fact that a stable complex was formed between the ligands and the protein. More precisely, stable RMSD and RMSF indicate that the inhibitors retained their consistent binding pose with a strong interaction towards the target enzyme. The most persistent interactions are formed via hydrogen bonds, water bridges, and ionic bonds, particularly with residues Arg-108, Lys-193, Gly-255, and Gly-257 of KHK. Hydrogen-bonding interactions between ligand and protein were observed with these amino acids, confirming the initial docking results that further support the reliability of the binding mode predicted during the docking studies. At 40 and 100 ns, 1 showed fast dynamics of interaction; most of them are insignificant for the overall stability of the complex. The study thus shows that the important residues and interactions were stable along the simulation frames and supported the binding pose. Results from MD simulations provided information about stability and interactions in protein–ligand complexes, confirming the reliability of the hits selected in the current study [69]. The presence of the carboxylic acid group in all five hits and PF-06835919 emphasizes its important role in the binding interaction of KHK inhibitors, contributing to the stability and affinity of the inhibitor–enzyme complex. This is interesting because, in fact, the carboxylic acid group can hydrogen bond with active site residues such as Gly-255 and Gly-257 to help maintain the inhibitor in the correct orientation and stabilize the inhibitor. Interaction with Arg-108, along with other positively charged residues, enhances the binding strength and specificity of the inhibitors [70]. While traditional virtual screening workflows typically apply shape similarity early in the process, we used shape similarity screening in a post-screening manner to refine and validate that the final shortlisted compounds would not only meet our docking and binding affinity criteria but also mimic some of the desired structural features of the reference compound. This will add a layer of confirmation that the hits possess the desired shape attributes and, therefore, their potential to act as effective inhibitors [49].

Arg-108 is in the ATP-binding site of the KHK enzyme, and it maintains Coulombic interactions to stabilize inhibitor or substrate binding. An acidic or hydroxyl group can be introduced to form favorable interactions with Arg-108 and can render the compound more inhibitory with higher affinity, as evidenced in the binding mode of PF-06835919. Gly-255 and 257 residues form the backbone in the KHK binding pocket. Their small size permits flexibility in the binding pocket by forming hydrogen bonds with inhibitors, resulting in improved complex stability and enhanced binding affinity. Cys-282 is located proximal to the ATP-binding site. It was shown to interact directly with the binding pocket and contribute to the overall conformation of the enzyme. Potential inhibitors, including the pyrimidinopyrimidine series, were reported to form hydrogen bonding interactions with the backbone CO of Cys-282. This residue is involved in the proper shape and conformation of the active site; therefore, mutations to alanine may result in major changes in binding affinity. The observed trends in docking scores across these mutations highlight the functional significance of these residues in modulating the binding affinity of the potential 5 hits [32].

ADMET predictions were conducted post-MD simulations to provide more logical interpretability as they were applied to compounds with confirmed stability in their protein–ligand complexes. Therefore, this approach reduced the likelihood of advancing false positives to the ADMET screening stage. The compounds displayed favorable pharmacokinetic properties, as evaluated through ADMET profiling. All compounds showed excellent intestinal absorption except for **5**, which required structural optimization to enhance its absorption. None of the compounds were predicted to penetrate the blood–brain barrier (BBB) or central nervous system (CNS), indicating limited distribution via these pathways, which is advantageous for targeting liver-specific metabolic diseases. Metabolism studies revealed that the selected compounds do not inhibit CYP450 enzymes, suggesting they are metabolized through alternative pathways, thereby reducing the risk of drug–drug interactions and bioaccumulation. Excretion profiles indicated efficient clearance from the body [71].

The evaluation of drug-likeness properties using Lipinski’s rule of five and other criteria confirmed that the compounds generally adhere to established drug-like profiles. Medicinal chemistry assessments indicated good synthetic accessibility and no significant alerts for pans assay interference structures (PAINS). Of the five hits, **4** demonstrated the most promising characteristics, reinforcing its potential as a drug candidate for KHK [72,73,74].

## 4. Materials and Methods

### 4.1. Experimental Design

The Comprehensive Marine Natural Products database (CMNPD) was retrieved from the online serve (https://www.cmnpd.org/, accessed on 25 June 2024) [75], which is a public web-accessible database containing over 46,000 compounds. Around 47,000 compounds derived from bacteria, fungi, and algae were initially collected in an sdf file format and subsequently processed using the online FAFDrugs4 server [76]. This processing involved a sequential application of drug-like soft filters, toxicophore screening [76], and Eli Lilly MedChem rules, which systematically removed compounds deemed non-drug-like, toxic [77,78], or problematic (PAINS compounds) from the dataset [79], resulting in 12,470 compounds—followed by preparation using the LigPrep tool in Maestro to add missing hydrogen atoms, build and energetically minimize outputs, optimize with the OPLS3e force field, convert to their respective 3D chemical structures, and ionize stereoisomers at a neutral pH of 7.0 ± 2.0 using Epik. The generation of tautomer and desalt was checked, and the stereoisomers were left to contain specific chirality to produce 32 isomers per ligand at most, and energy was minimized for molecular modeling [80]. The filtered 12,470 compounds were subjected to high throughput virtual screening (HTVS) using Glide software within Schrödinger. This helped rapidly assess the binding potential of each compound against the target protein, Ketohexokinase-6W0Z. A total of 9032 compounds with favorable GlideScore passed this initial screening; these were then subjected to a more stringently filtered Standard Precision (SP) docking. This includes a more refined binding pose and energy calculation considering extra molecular interactions. The screening at this stage narrowed the pool further to 1053 compounds with improved docking scores and plausible binding poses within the active site of the target protein. The top 1053 compounds passed through an additional filter of Extra Precision Docking that provides a highly accurate assessment of binding interactions, considering both ligand and receptor side chains. This resulted in XP docking identifying 279 compounds with the most favorable binding pose and the best docking score, hence being strong potential inhibitors. To further refine this selection, the top 279 compounds were subjected to MM-GBSA calculations. This method estimates the free binding energy of the ligand–protein complexes, therefore making more reliable predictions for the binding affinity of a compound. Indeed, from these calculations, the final 5 hits had the lowest binding free energies and stable binding conformation, as shown in Figure 11.

### 4.2. Retrieval of Ketohexokinase-c Crystal Structure, Protein Reliability, and Preparation for Docking Analysis

The X-ray crystallographic structure of KHK-c from a human complexed with PF-06835919 (PDB ID: 6W0Z) [32] was obtained from the Protein Data Bank (https://www.rcsb.org/, accessed on 25 June 2024) as displayed in Figure 12. All 3D structural figures were generated using Pymol [81]. A structural analysis panel was utilized to obtain a protein reliability report to assess the reliability of retrieved structures [82]. The Protein Preparation Wizard in Maestro molecular modeling software, 13.6, was used to repair missing residues and side chains, protonate histidine residues using PROPKA at pH 7.0, fill loops with Prime, and assign bond orders. The Het (hetero atom) state was created for PF-06835919 (S6D 301) at the protein’s active site during preprocessing. Polar hydrogens were added, non-essential water molecules were removed, and all heavy atoms converged to a root mean square deviation (RMSD) of 0.3 Å. Performing an independent Ramachandran plot analysis is essential for verifying the structural integrity of key residues, especially those falling within or near the active site or binding pocket. This is a critical step since the quality of the crystal structure will define the quality and reliability of further computational simulations and analyses, such as molecular docking and dynamic simulations. To assess the general quality of the minimized structure, the Ramachandran plot was generated and compared with the unminimized. The entire structure was minimized and optimized with the OPLS3 force field and was subsequently used for the docking process [82].

### 4.3. Binding Site Determination and Docking Validation

A docking grid was generated using the receptor grid generation tool to specify the 3D coordinates of the active sites of ketohexokinase (KHK), set at (−4.49, 0.93, 18.79) for (x, y, z) within a confined volume of 20 Å. This setup created a centroid in the receptor’s active site and established a grid box. The co-crystallized ligand was redocked to ensure accuracy, and docking poses and interactions were validated through Maestro’s structure superimposition and RMSD alignment calculations [52,83,84]. Using a van der Waals (vdW) radius scaling factor of 1.00 and a partial charge cutoff of 0.25, the receptor grid was centered on the bound ligand. The binding site was enclosed within the grid box, adhering to default parameters and without applying any constraints. Following this setup, the docking process was repeated and validated across three screening settings.

### 4.4. Non-Covalent Docking Screening (Semi-Rigid Docking)

The ligand was docked using the Glide tool without constraints [85], employing a van der Waals (vdw) radius scaling factor of 0.80 and a partial charge cut-off of 0.15. The ligands’ flexibility was considered, while the protein was considered a rigid structure, with all other parameters set to their default values. GlideScore, implemented in Glide, was utilized to predict binding affinity and rank ligands. The Pose Rank was utilized to identify the optimal docking pose for each ligand. Subsequently, the compounds were thoroughly examined based on their binding scores and a detailed analysis of all binding interactions.

### 4.5. Induced Fit Docking (Flexible Docking)

The induced-fit docking (IFD) technique, developed by Schrödinger, is used to model how ligand binding induces conformational changes [86]. This method involves several steps, as outlined in [46], for docking one or more ligands. Using the IFD tool in Maestro, each ligand undergoes initial docking using a softened potential (van der Waals radii scaling) and flexible conformational sampling. After that, side-chain prediction is conducted within a specified distance of each ligand pose. Following this, residues and the ligand in each protein/ligand complex pose undergo minimization. Finally, a favorable binding pose is predicted based on the IFD score.

### 4.6. Molecular Mechanics-Based Re-Scoring

The binding complexes were re-scored using molecular mechanics generalized Born surface area (MM/GBSA) docking to improve the accuracy of affinity predictions [87].
ΔG binding free energy = ΔG binding, vacuum + ΔG solvation, complex − (ΔG solvation, ligand + ΔG solvation, receptor).

MM/GBSA enhances accuracy by allowing both the ligand and receptor to remain flexible, which is critical for physiological relevance [88]. Therefore, an intensive MM/GBSA simulation was used to rank the binding affinities of the eight identified hits against the KHK target. To achieve this, the initial extensive precision (XP) complexes of KHK hits and PF-06835919 underwent MM/GBSA docking in Maestro. Flexibility was incorporated by adjusting the distance between the hits or PF-06835919 and KHK to 5 Å. The simulation employed the VGSB solvation model alongside the OPLS3 force field [88].

### 4.7. Molecular Dynamics Simulation Studies

The high scoring complex was further assessed through molecular dynamics (MD) simulations in Maestro 13.6 software. The simulation system was prepared using the Desmond System Builder where the receptor–ligand complex was immersed in the TIP3P solvent model with a buffer system containing 0.15M sodium chloride [89]. The solvent box volume was minimized under the OPLS4 force field, resulting in the final system containing approximately 25,000 atoms. The final system was retrieved for a 100 ns MD simulation, beginning with the system relaxation using the default protocol. The isothermal-isobaric NPT entity at 310 K temperature and 1.103 bar pressure was adopted during the simulation [89,90]. Other parameters such as Coulombic cutoff distance and a reversible reference system propagator algorithm (RESPA) integrator were kept at default settings. Following the simulation, the Desmond Simulation Interaction Diagram tool was used to analyze the results [91,92,93].

### 4.8. Shape-Based Screen

The 2023 version of Schrödinger’s Shape Screening tool was employed for shape screening. PF-06835919 served as the reference structure during this process. Six compounds underwent screening utilizing the pharmacophore volume scoring technique, which evaluates each compound as an assembly of pharmacophore features, including aromatic groups, hydrogen bond acceptors (HBAs), hydrogen bond donors (HBDs), hydrophobic regions, as well as positively and negatively charged groups. The shape similarity score (Shape Sim score) was derived from the most matching features among these compounds [49].

### 4.9. In Silico Site-Directed Mutagenesis

The residues involved in consistent interactions and stability of the complexes were chosen for alanine scanning analysis. Alanine scanning was performed ne in the Maestro BioLuminate 4.4.123 package. These key amino acid residues were selected according to the docking interactions and residue analysis and were manually mutated to ALA. The XP calculation in the Glide tool was followed with the intention of using the generated docking poses and scores for qualitative and quantitative analyses. The goal was to scan for variations in docking scores due to residues mutations [94].

### 4.10. ADMET Properties and Drug-Likeness Predictions

We utilized the pkCSM web server (http://biosig.unimelb.edu.au/pkcsm/prediction, accessed on 2 July 2024). Ref. [95] to predict descriptors for both ADMET (absorption, distribution, metabolism, excretion, and toxicity) and drug-likeness properties of the final selected potential inhibitors. Eight molecular descriptors were generated to characterize the ADMET properties of the potential KHK hits. Additionally, physicochemical properties, medicinal chemistry, and drug-likeness were applied via SwissADME (http://www.swissadme.ch/, accessed on 5 July 2024) [96].

## 5. Conclusions

This study successfully identified several promising KHK inhibitors, including **1** (CMNPD12445), **2** (CMNPD799), **3** (CMNPD24755), **4** (CMNPD27745), and **5** (CMNPD21775) with strong binding affinities and favorable ADMET profiles. The integration of SBDD and LBDD methodologies provides an all-encompassing approach toward discovering novel inhibitors targeting the key player enzyme, KHK, in fructose metabolism. Our findings demonstrated that compounds identified from marine sources had a highly effective potential for therapeutic agent development against fructose-related metabolic disorders, including NAFLD and obesity.

The stability of these compounds bound to the active site of KHK was further validated by computational analyses such as molecular docking, induced-fit docking, MM/GBSA calculations, and MD simulations. Of these, compound 1 and compound 2 exhibited robust binding interactions with the important amino acids lining the KHK binding pocket, which remained stable throughout the period of the MD simulation. These compounds were also computationally found to exhibit favorable pharmacokinetic properties, which supports the prospect of these compounds acting as drug candidates.

The significance of this study lies in its contribution to the growing body of knowledge on marine-derived natural products as a drug for metabolic diseases. Some of the identified KHK inhibitors could thus form the basis for developing novel therapeutic approaches against the undesirable aspects of excessive fructose intake. Further experimental validation and optimization may, thus, be required to confirm their efficacy and safety.

## Figures and Tables

**Figure 1 marinedrugs-22-00455-f001:**
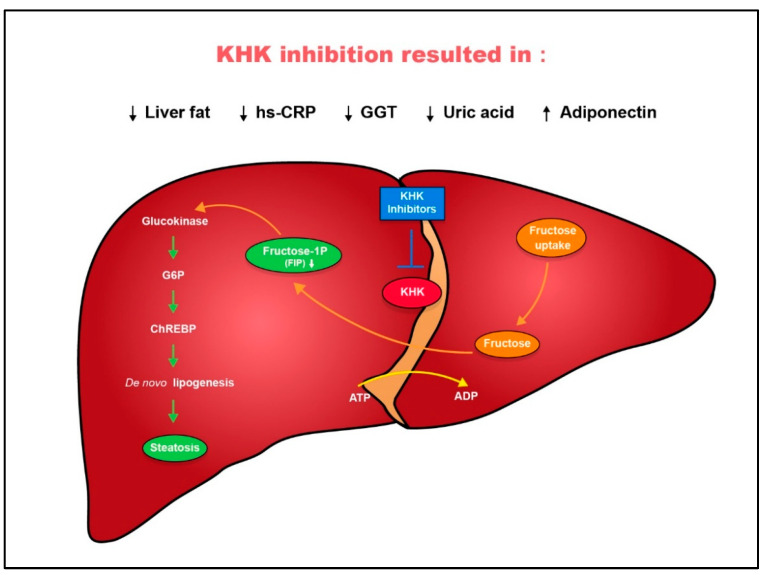
Inhibition of KHK reduced liver fat in adults with non-alcoholic fatty liver disease (NAFLD).

**Figure 2 marinedrugs-22-00455-f002:**
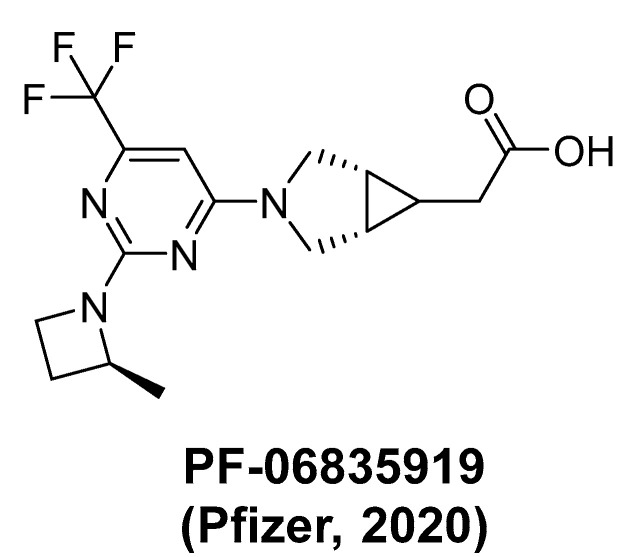
KHK inhibitor.

**Figure 3 marinedrugs-22-00455-f003:**
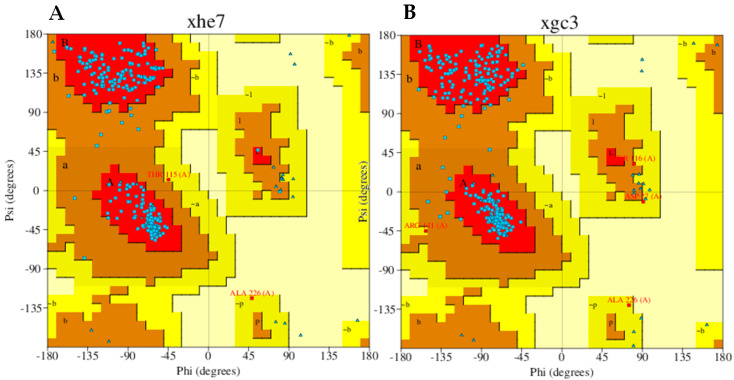
Ramachandran plot. (**A**). Energy Pre-minimized KHK. (**B**). Energy minimized KHK. Details about the coloring of the plot can be interpreted from the PDBSum generated server.

**Figure 4 marinedrugs-22-00455-f004:**
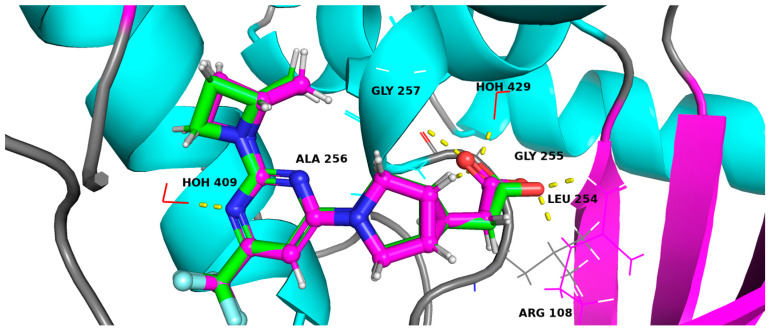
Comparison of binding poses of the co-crystallized ligand (magenta) and the redocked ligand (green) within the KHK binding site, with an RMSD of 0.3004 Å.

**Figure 5 marinedrugs-22-00455-f005:**
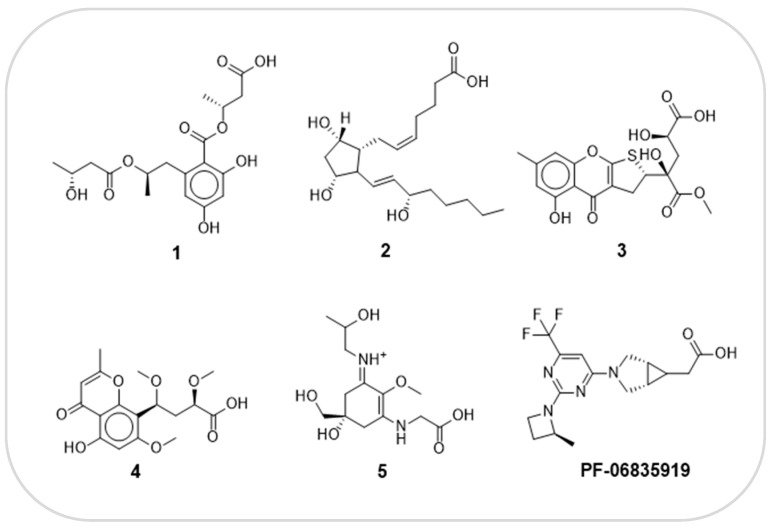
The 2D structures of PF-06835919 and the five promising hits.

**Figure 6 marinedrugs-22-00455-f006:**
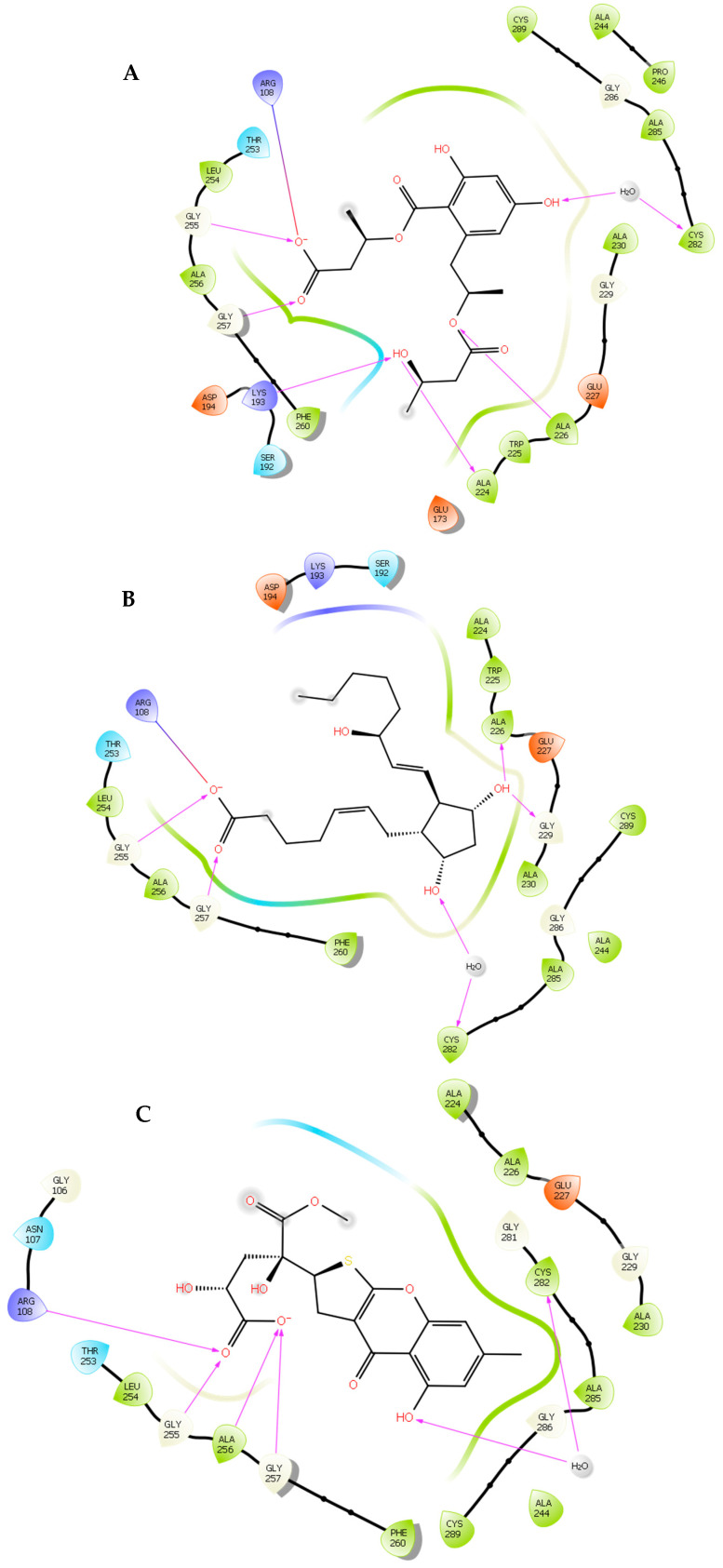
Two-dimensional ligand–protein binding interactions of KHK bounded to the top five hit candidates and control. (**A**). Compound 1. (**B**). Compound 2. (**C**). Compound 3. (**D**). Compound 4. (**E**). Compound 5. (**F**). PF-06835919. 
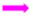
 = Hydrogen bond interactions, 
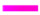
 = Ionic interactions.

**Figure 7 marinedrugs-22-00455-f007:**
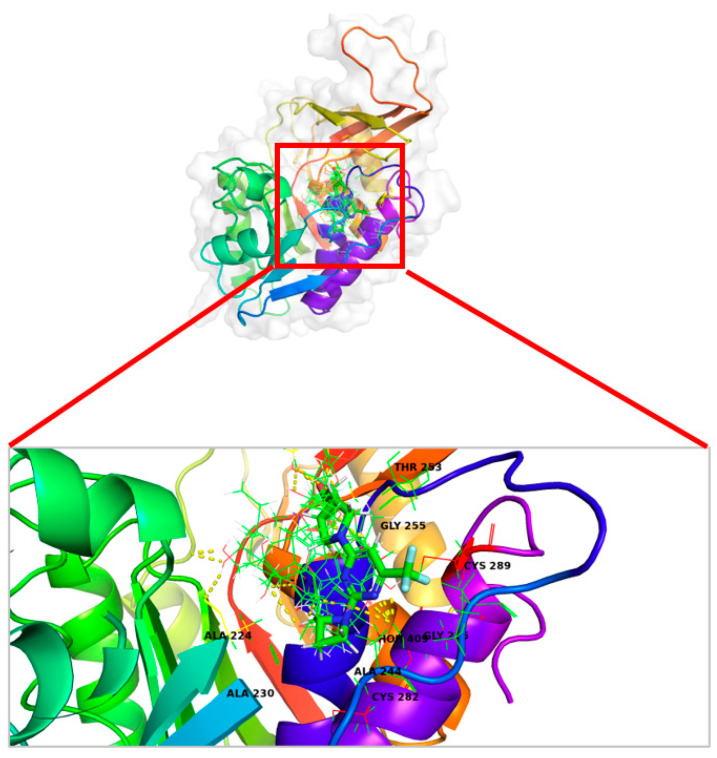
Ligand–protein binding interactions. A. Overall view of three-dimensional cartoon and surface representation of KHK bounded to the top five hit candidates (wire, green) and PF-06835919 (thin tube, atom color) (PDB entry: 6W0Z). B. Focused view of the catalytic chamber showing the binding pose of the top five hit candidates (wire, green) and PF-06835919 (thin tube, atom color).

**Figure 8 marinedrugs-22-00455-f008:**
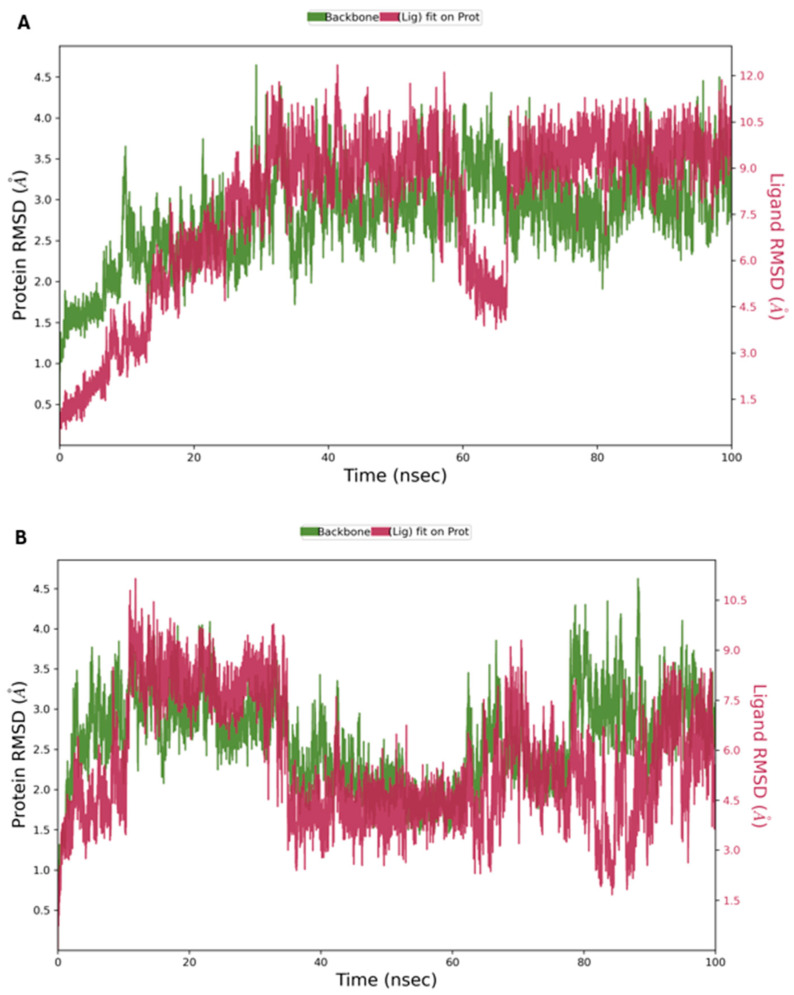
The equilibration of conformational changes in KHK in complex with **1** (**A**) and **2** (**B**) during the 100 ns analysis.

**Figure 9 marinedrugs-22-00455-f009:**
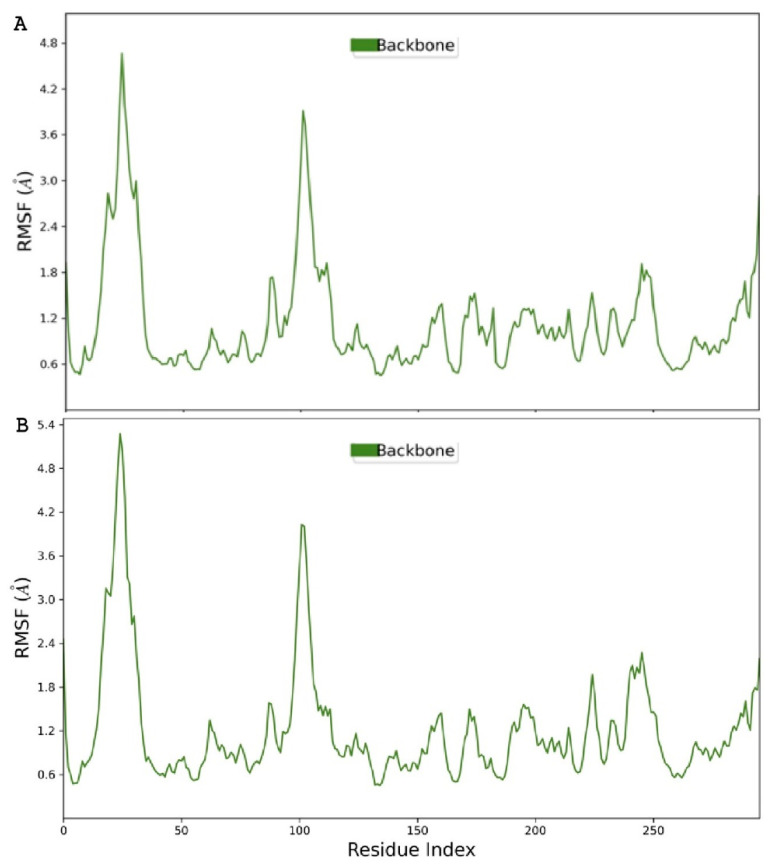
The comparable conformational changes of KHK in complexes with **1** (**A**) and **2** (**B**) during the 100 ns MD analysis.

**Figure 10 marinedrugs-22-00455-f010:**
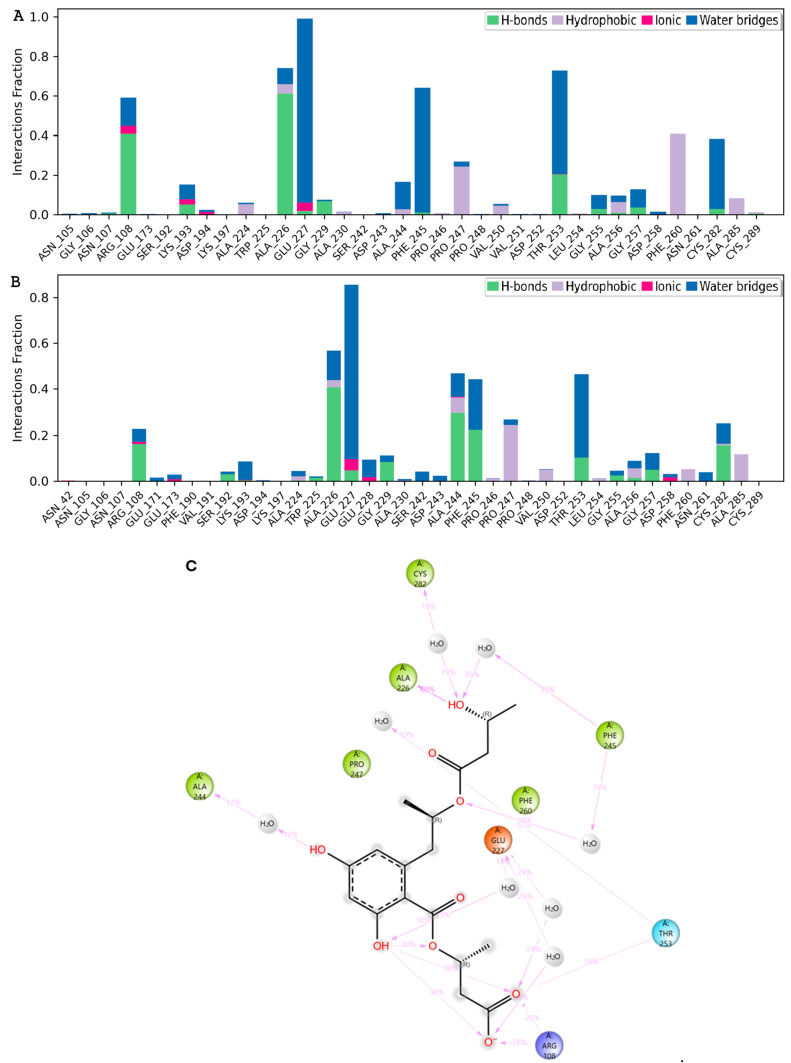
The constant interactions of KHK with **1** (**A**) and **2** (**B**) and their absolute percentage contributions to the respective final binding conformations (**C**,**D**).

**Figure 11 marinedrugs-22-00455-f011:**
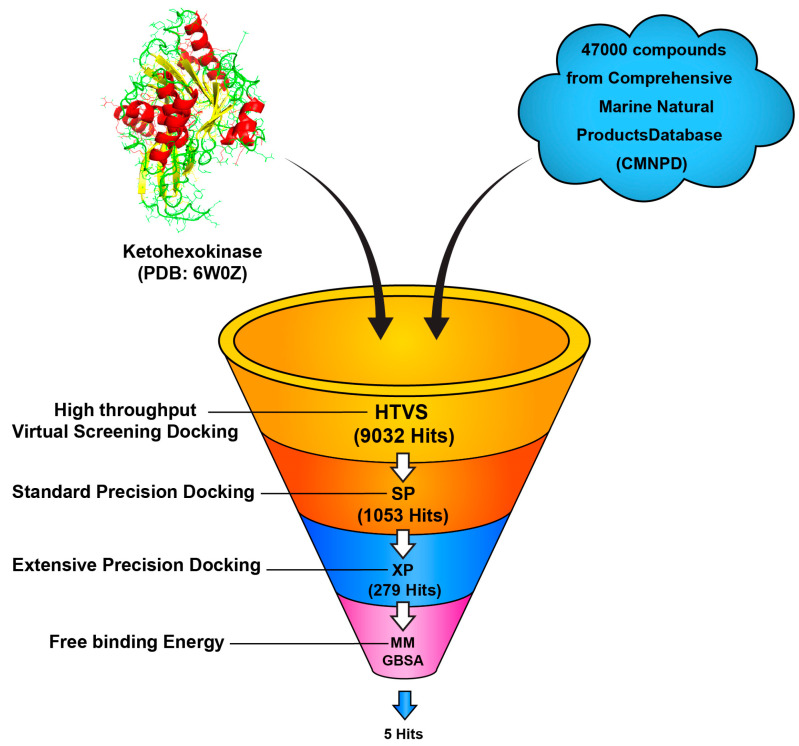
Workflow of the virtual screening process for identifying potential ketohexokinase inhibitors.

**Figure 12 marinedrugs-22-00455-f012:**
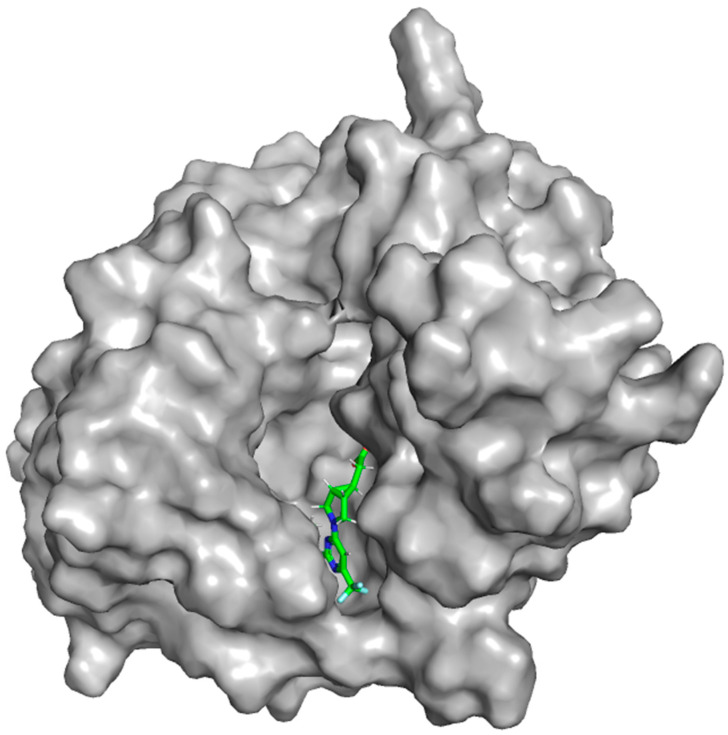
Crystal structure of ketohexokinase (KHK) bound to PF-06835919 (S6D 301).

**Table 1 marinedrugs-22-00455-t001:** Docking scores and induced-fit docking of the five hits at the binding site of KHK (PDB entry: 6W0Z).

Compound	Name/ Source	Glide Score Docking (Semi-Rigid) ^a^	Induced-Fit Docking (IFD) (Flexible) ^a^	Ionic Interactions (Semi-Rigid)	H-Bond Interactions (Semi-Rigid)
**1**	NA/ Halorosellinia oceanica	−10.51	−10.00	Arg-108	Lys-193
					Ala-224
					Ala-226
					Gly-255
					Gly-257
					H_2_O-409(Bridge with Cys-282)
**2**	PGF2α ^†^/Hydropuntia Edulis	−9.47	−9.69	Arg-108	Ala-226
					Gly-229
					Gly-255
					Gly-257
					H_2_O-409(Bridge with Cys-282)
**3**	Oxalicumone D/Penicillium oxalicum	−9.44	−9.28		Arg-108
					Gly-255
					Ala-256
					Gly-257
					H_2_O-409(Bridge with Cys-282)
**4**	Rhytidchromone E/ Rhytidhysteron rufulum	−9.28	−9.13		Arg-108
					Gly-255
					Gly-257
					H_2_O-409(Bridge with Cys-282)
**5**	Aplysiapalythine A/Aplysia californica	−9.23	−8.28	Arg-108	Lys-193
					Ala-226
					Gly-255
					Gly-257
					H_2_O-409(Bridge with Cys-282)
**PF-06835919**	(Control)	−10.73	−10.69	Arg-108	Gly-255
					Gly-257
					H_2_O-409(Bridge with Cys-282)
					H_2_O-429

^a^ Higher negative value indicate higher binding interactions within the binding pocket.

**Table 2 marinedrugs-22-00455-t002:** MM-GBSA net binding energy of the compounds/control.

Compound	ΔG Binding (kcal/mol) ^a^	ΔG Binding H-bond	ΔG Binding vdW	ΔG Binding Solve GB
**1**	−51.66	−6.34	−43.41	−7.53
**2**	−45.23	−5.9	−42.75	−17.64
**5**	−38.29	−6.09	−34.53	−3.27
**3**	−35.66	−4.09	−40.55	−1.08
**4**	−31.25	−4.57	−34.9	−3.97
**PF-06835919 (Control)**	−92.35	−6.18	−46.85	−0.34

^a^ Higher negative value indicate a higher binding affinity within the binding pocket.

**Table 3 marinedrugs-22-00455-t003:** Shape similarity of the hits and control.

Compound	Shape Similarity ^b^
**2**	0.424
**5**	0.32
**4**	0.319
**1**	0.289
**3**	0.267
**PF-06835919 (Control)**	1

^b^ Values closer to 1 indicate higher shape similarity to the control. Cutoff score ≥0.4.

**Table 4 marinedrugs-22-00455-t004:** Mutated residues and docking scores of potential hits.

Residue Mutated to Alanine	1	2	3	4	5
R108A	−7.55	−7.36	−8.97	−6.99	−6.42
G255A	−7.81	−7.13	−9.03	−7.42	−6.30
G257A	−7.44	−7.50	−8.57	−7.77	−6.71
C282A	−8.66	−8.71	−11.82	−9.33	−7.49

**Table 5 marinedrugs-22-00455-t005:** ADMET profiling of the five promising drug candidates.

ADMET Parameters	PF-06835919 (Control)	1	2	3	4	5
**Absorption**						
Water solubility (log mol/L)	−3.117	−2.909	−3.594	−3.471	−3.21	−0.90
Caco2 permeability (log Papp in 10^−6^ cm/s)	1.045	−0.55	0.39	−0.31	0.07	−0.32
Intestinal absorption (human) (% Absorbed)	95.07	40.97	48.87	46.84	59.53	17.61
P-glycoprotein substrate (Yes/No)	No	Yes	Yes	Yes	Yes	Yes
**Distribution**						
BBB permeability (log BB)	−0.91	−1.7	−1.03	−1.37	−1.01	−0.93
CNS permeability (log PS)	−3.04	−3.82	−3.24	−3.67	−3.39	−4.39
Metabolism						
CYP2D6 substrate (Yes/No)	No	No	No	No	No	No
CYP3A4 substrate (Yes/No)	Yes	Yes	No	Yes	Yes	No
CYP1A2 inhibitior (Yes/No)	No	No	No	No	No	No
CYP2C19 inhibitior (Yes/No)	No	No	No	No	No	No
CYP2C9 inhibitior (Yes/No)	No	No	No	No	No	No
CYP2D6 inhibitior (Yes/No)	No	No	No	No	No	No
CYP3A4 inhibitior (Yes/No)	No	No	No	No	No	No
**Excretion**						
Total Clearance (log ml/min/kg)	−0.13	1.11	1.55	0.14	0.88	0.67
Renal OCT2 substrate (Yes/No)	No	No	No	No	No	No
**Toxicity**						
AMES toxicity (Yes/No)	No	No	No	Yes	No	No
Max. tolerated dose (human) (log mg/kg/day)	0.60	0.68	0.59	0.48	0.61	0.50
hERG I inhibitor (Yes/No)	No	No	No	No	No	No
Hepatotoxicity (Yes/No)	Yes	No	No	Yes	Yes	Yes

**Table 6 marinedrugs-22-00455-t006:** Physicochemical properties, drug-likeness, and medicinal chemistry prediction of top five promising hits.

Molecule Properties	PF-06835919 (Control)	1	2	3	4	5
**Physicochemical properties**						
Molecular Weight	356.34	384.38	354.48	410.4	352.33	303.33
LogP	2.25	1.37	3.04	0.56	1.2	−3.06
#Acceptors	5	8	4	9	8	6
#Donors	1	5	4	4	2	6
#Heavy atoms	25	27	25	28	25	21
#Arom. heavy atoms	6	6	0	10	10	0
Fraction Csp3	0.69	0.50	0.75	0.39	0.41	0.69
#Rotatable bonds	5	11	12	6	7	7
Molar refractivity	90.40	94.22	100.45	98.31	88.91	75.55
TPSA (Å^2^)	69.56	150.59	97.99	179.80	115.43	133.22
**Drug-likeness**						
Lipinski alert	Pass	Pass	Pass	Pass	Pass	Pass; 1 violation: #Donors > 5
Ghose	Pass	Pass	Pass	Pass	Pass	No; 1 violation: WLOGP < −0.4
Veber	Pass	No; 2 violations: Rotors>10, TPSA > 140	No; 1 violation: Rotors>10	No; 1 violation: TPSA > 140	Pass	Pass
Egan	Pass	No; 1 violation: TPSA > 131.6	Pass	No; 1 violation: TPSA > 131.6	Pass	No; 1 violation: TPSA > 131.6
Muegge	Pass	No; 1 violation: TPSA > 150	Pass	No; 1 violation: TPSA > 150	Pass	No; 1 violation: H-don > 5
Bioavailability Score	0.85	0.11	0.56	0.11	0.56	0.55
**Medicinal chemistry**						
PAINS	Pass	Pass	Pass	Pass	Pass	Pass
Brenk	Pass	1 alert: more_than_2_esters	1 alert: isolated_alkene	Pass	Pass	1alert: imine_1
Synthetic accessibility	3.91	4.19	5.04	4.77	4.20	4.56

## Data Availability

All data generated or analyzed during this study are included in this published article.

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
