# Peer review of "Exploring Marine-Derived Compounds: In Silico Discovery of Selective Ketohexokinase (KHK) Inhibitors for Metabolic Disease Therapy"

_marinedrugs, 2024, doi:10.3390/md22100455_

Round 1
Reviewer 1 Report
Comments and Suggestions for Authors
The manuscript titled "Exploring Marine-Derived Compounds: In Silico Discovery of Selective Ketohexokinase (KHK) Inhibitors for Metabolic Disease Therapy" presents a well-executed study on selective KHK inhibitors using in silico methods, effectively employing both structure-based and ligand-based drug design approaches. While the manuscript is well-written and emphasizes the significance of KHK as a therapeutic target, several points need to be addressed, particularly in the results/discussion and methodology sections, to meet the standards of Marine Drugs. Given these considerations, I recommend a major revision to enhance the manuscript's clarity and ensure alignment with field standards.
1. Why did the authors perform Ramachandran plot analysis on the crystallographic structure? This approach is not typically applied to experimental structures, as they are validated before being included in the Protein Data Bank. The rationale for choosing this analysis should be clearly explained by the authors.
2. In Figure 4, the authors are requested to revise the image. I recommend using PyMOL or Chimera software to generate a high-quality (>300 DPI) three-dimensional representation of the protein structures. Additionally, the interaction profile is difficult for readers to understand, as the interacting residues in the protein are not labeled. Lastly, each type of atom should be colored appropriately for clarity.
3. Authors are requested to include an appropriate legend in Figure 6, clarifying the color-scheme used for the interactions.
4. Figure 7 should be revised, as the interaction profile is difficult to visualize and understand. A zoomed-in view may improve clarity and help readers better interpret the interactions displayed. Additionally, it is important to color the ligand according to the type of atom for improved distinction and readability.
5. In Table 1, the authors are requested to clarify which type of docking (semi-rigid or induced-fit) was used to provide the interactions shown. This distinction should be made explicit.
6. The experimental design of the study was not clearly explained in the manuscript. It is unclear how the authors identified the 5 hits from a database containing approximately 50,000 ligands. The authors mentioned filtering molecules based on drug-like properties and toxicophores, resulting in 12,470 compounds, but the subsequent selection of only 5 hits through XP docking requires further clarification. I recommend providing a detailed description of the virtual screening design and including a flowchart to enhance clarity, displaying all steps involved and the number of compounds remaining in the workflow after each step.
7. Given the computationally intensive nature of molecular dynamics (MD) simulations, why did the authors conduct molecular dynamics simulations before other simpler screening processes, such as ADMET prediction? The rationale for this decision should be clearly explained.
8. In Figure 8, the authors are requested to provide a plot with a more proportional height to avoid flattening the graph. The image needs to clearly indicate whether a plateau in the RMSD values was established, as this would demonstrate that the trajectory is in equilibrium, justifying subsequent analyses. As it is currently presented, this behavior cannot be clearly assessed. It is important to note that representative results from molecular dynamics simulations can only be drawn during the equilibration phase. Therefore, all conclusions should be based solely on data collected within this period to ensure accuracy and reliability. Did the authors consider only the equilibrated period for their analysis? It should be clarified.
9. A proper discussion of the MD simulations should be provided in the manuscript, clearly explaining the RMSD and RMSF analysis and their implications.
10. Figure 10 (C and D) should be revised, providing a clear and high-quality image of the interactions.
11. Shape similarity is typically applied during the virtual screening workflow, not afterward. The authors are requested to explain their decision to use it after the initial screening, specifically to evaluate only 5 compounds. Additionally, the criteria for the shape similarity score should be clarified, including which score can be considered a hit.
12. In the alanine scanning analysis (section 2.7), the authors should discuss the roles of the identified residues in conjunction with their results. This discussion should include how each residue contributes to the stability and function of the protein, as well as its role in the protein's inhibition or activity. Understanding the implications of these substitutions on the interaction profile will enhance the overall interpretation of the results and clarify the significance of these residues in the context of the study.
13. The Conclusion section requires improvement. It should emphasize the main findings of the study and the significance of the work's contributions.
Author Response
Please see the attachment. Thank you so much for taking the time to review my manuscript.

Reviewer 2 Report
Comments and Suggestions for Authors
The paper submitted by Alturki for publication on Marine drugs describes the computational prediction and characteristics of several drug candidates active against ketohexokinase (KHK) enzyme. Five compunds have been selected for their promising activity and their computational properties have been predicted. Docking and other in silico studies have been performed to reinforce their potential application in metabolic disorders.
However, there are several points that require attention to improve the discussion of the results and the clarity of the aims of this paper. In particular, in many points of the paper is not clear which compound the author is referring to. This should be clarified and corrected since it is quite difficult understand the workflow. I suggest also to replace the code number of selected compounds with numbers (1-5).
Line 75: “Given the time-consuming and costly nature of traditional drug discovery processes”. This sentence should be better explained. Proposal: “The main hurdle to using natural products, especially those of marine origin, is their reduced availability as well as complicated extraction and purification procedures. One example is avarone, a promising sesquiterpene quinone, which despite being isolated in good quantities from its original matrix deserves further biotechnological development”. Refer to 10.3390/pharmaceutics15020528 that should be cited at that level considering also the computational studies performed on this compound and the aim of this paper.
Line 90: structure-based drug design (SBDD) and ligand-based drug design (LBDD). For a clearer discussion, I suggest to better describe this two drug discovery processes.
Line 92: what compounds?
Line 120: 0% should be modified in no residues
Figure 6C add a higher resolution structure. The same for the arrows in figure 10 that are of low quality.
After these revisions, the article may be accepted for publication.
Comments on the Quality of English Language
The English language should be revised; the sentences should be more concise and minor types should be addressed.
Author Response

(The authors gave the same response as above.)

Round 2
Reviewer 1 Report
Comments and Suggestions for Authors
The revised version of the manuscript has significantly improved the quality and clarity of the results presentation. The authors have effectively addressed the previous concerns by providing a more detailed and structured explanation of the in silico methods used, as well as a clearer visualization of the virtual screening workflow. The justification for the selection criteria of potential inhibitors is now well articulated, and the discussion of the findings is more coherent. I believe this revised manuscript meets the standards of Marine Drugs and is suitable for publication.
Reviewer 2 Report
Comments and Suggestions for Authors
The author addresses all the suggestions of the first round of revision and currently the manuscript can be ccepted for publication